# Adsorption of Cadmium by *Brassica juncea* (L.) Czern. and *Brassica pekinensis* (Lour.) Rupr in Pot Experiment

Rongrong Ying [1,†], Bing Xia [1,2,†], Xiaowen Zeng [3], Rongliang Qiu [4,5], Yetao Tang [4,5,*] and Zhewei Hu [1,*]

1. Nanjing Institute of Environmental Sciences, MEE, Nanjing 210042, China; yingrongrong@nies.org (R.Y.); sweetblues@gmail.com (B.X.)
2. Anhui Academy of Environmental Science Research, Hefei 230071, China
3. School of Public Health, Sun Yat-sen University, Guangzhou 510275, China; zxw63@mail.edu.cn
4. School of Environmental Science and Engineering, Sun Yat-sen University, Guangzhou 510275, China; qiurl@scau.edu.cn
5. Guangdong Provincial Key Lab of Environmental Pollution Control and Remediation Technology, Guangzhou 510275, China
* Correspondence: Tangeestyt@mail.sysu.edu.cn (Y.T.); huzhewei@nies.org (Z.H.)
† These authors contributed equally to this work.

**Abstract:** Heavy metal pollution in farmland threatens human life. It is not clear whether crops can adsorb heavy metals. In this study, the cadmium accumulation and tolerance in Chinese cabbage *Brassica pekinensis* (cv. Xiaoza-56) and the known Cd-accumulator *Brassica juncea* in hydroponics and pot experiment were investigated. Furthermore, we evaluated their potential on the phytoextraction of Cd-contaminated soil. The hydroponics with 1–50 μM Cd concentrations showed that both *B. juncea* and *B. pekinensis* had high Cd accumulation and tolerance with translocation factor closed to 1 at Cd levels <25 μM. The pot study conducted with 5 to 100 mg Cd kg$^{-1}$ soil indicated that *B. juncea* showed less tolerance and accumulation to Cd than *B. pekinensis*, especially at higher Cd levels. The bioconcentration factor was much higher than 1 in both *B. juncea* and *B. pekinensis* grown in <40 mg Cd kg$^{-1}$ soil without showing biomass reduction. In the model evaluation, the ability of *B. juncea* and *B. pekinensis* to reduce the initial soil Cd concentration of 20 and 5 mg kg$^{-1}$ to specific targets with a lower or higher biomass of 4 or 20 t ha$^{-1}$, respectively. The above results indicate that *B. juncea* and *B. pekinensis* (cv. Xiaoza-56), which the latter is a better candidate for Cd phytoextraction in moderated Cd-contaminated soil. The results provide a reference for Cd pollution control.

**Keywords:** *Brassica pekinensis*; *Brassica juncea*; Cd; phytoextraction

## 1. Introduction

Agricultural soil pollution with toxic metals is of increasing concern due to food safety issues and potential health risks [1]. Heavy metals are metals with a density greater than 4.5 g/cm$^3$, including gold, silver, mercury, lead and cadmium. Heavy metals enter the human body mainly through food and accumulate in the human body to a certain extent, which will cause chronic poisoning. Among the heavy metals, non-essential cadmium (Cd) is extremely toxic to most organisms. There was 13,000 ha of arable land in 25 regions in 11 provinces of China, have been polluted by Cd [2]. Since most of the efficient soil cleanup techniques are costly, and this kind of technology was difficult to popularize and apply [3]. New environmentally friendly methods based on low-cost technology are needed [4]. Plant adsorption of heavy metals is considered to be an effective method. Plant roots can absorb heavy metals from soil or water and transfer them to other parts so as to remove heavy metal pollution [5,6]. There are two main kinds of optional plants used in phytoextraction. One has relied on the use of plants known as hyperaccumulators, such as the Zn/Cd hyperaccumulator *Thlaspi caerulescens* [7–9] and *Arabidopsis halleri* L. [10], with a recognized ability of Zn or Cd phytoextraction. However, herbaceous plants grow slowly

and remain small in order to be preserved longer [3]. Moreover, this type of plant is rare, and only around 400 species have been found in the world. Consequently, the other option considered for soil remediation is using heavy-metal-tolerant plants with a fast growth rate or higher biomass [11]. Among them, the *Brassica* species has attracted a lot of attention in the last two decades because it has higher biomass production and grows faster compared to the well-known Zn/Cd hyperaccumulator *Thlaspi caerulescens* [12–14].

As a widely cultivated and widely sold vegetable, the annual sales of *Brassica juncea* in China was 1.5 billion yuan. *Brassica juncea* has been considered a plant with great potential for phytoremediation [10]. In our previous report, it was found that another *Brassica* species, *Brassica pekinensis*. R (Lour.) (cv. Xiaoza-56), could accumulate 20–75 mg kg$^{-1}$ of Cd (D.W) in the shoots. However, it did not show any phytotoxicity when grown in a vegetable garden with 1 mg kg$^{-1}$ of Cd soil [15]. Moreover, *B. pekinensis* (cv. Xiaoza-56) is one cultivar of Chinese cabbage with a short growth cycle and high biomass. This finding provided us a good native material that could be used for potential phytoextraction in Cd-contaminated soil. Conversely, whether *B. pekinensis* can adsorbed heavy metals remains unclear.

This paper contributes to the discussion on the potential Cd phytoextraction in two *Brassica* species, *B. juncea* and *B. pekinensis* (cv. Xiaoza-56). The experiment consisted of two parts. In the first part, the Cd-accumulation and Cd-tolerance of these two plants were compared by hydroponics. Second, the Cd extraction potential of the plants was investigated in a Cd-spiked pot experiment. In this study, based on the data obtained, the phytoextraction potential in moderate Cd-contaminated soil of *B. juncea* and *B. pekinensis* (cv. Xiaoza-56) was determined based on the model adopted from Zhao et al. [16]. The results provide a reference for heavy metal pollution control and ecological restoration.

## 2. Materials and Methods

### 2.1. Design of Hydroponic Experiment

In order to ensure maximum absorption of heavy metal ions by plants, seeds of *B. juncea* (from Agriculture Scientific Research Institute, Guangdong, China) and B. pekinensis (cv. Xiaoza-56 from the Dongsheng Hongjun plant nursery, Beijing, China) were sown in commercial garden soil. The seeds were cultured for three weeks and transferred to buckets containing 1.5 L of the Hoagland's solution. The Hoagland's solution includes 0.1 mM of KCl, 0.01 µM of $(NH_4)6Mo_7O_{24}$, 0.2 µM of $CuSO_4 \cdot 5H_2O$, 0.5 µM of $ZnSO_4 \cdot 7H_2O$, 0.5 µM of $MnSO_4 \cdot H_2O$, 10 µM of $H_3BO_3$, 0.7 mM of $K_2SO_4$, 2 mM of $Ca(NO_3)_2 \cdot 4H_2O$, 0.1 mM of $KH_2PO_4$ and 0.5 mM of $MgSO_4 \cdot 7H_2O$100 µM of Fe-EDTA. The solution was refreshed twice a week and aerated continually at the same time. We adjusted the pH of the solution to 5.8 with 1 M KOH. Growth conditions were controlled to an average temperature of $18 \pm 3$ °C at night and $25 \pm 3$ °C during the day. Six Cd treatments of 0, 1, 5, 10, 25 and 50 µM were set up by adding CdCl2·2.5 $H_2O$ to the Hoagland's solution acclimation after 1 week. There were 3 treatments and 2 seedlings in each treatment. After 21 days of exposure, plants were harvested, and the roots were also immersed in 0.02 M EDTA-Na$_2$ solution for 15 min to eliminate the $Cd^{2+}$ adsorbed at the root surface. Then, the plants were washed with water. The effect of Cd on plant growth was evaluated by measuring the shoot and root length as well as the plant's biomass (dry weight).

### 2.2. Design of Pot Experiment

In order to better simulate the adsorption of heavy metals by plants under natural environmental conditions, a pot culture experiment was conducted at seven soil levels (D.W) (5, 10, 20, 40, 60, 80 and 100 mg kg$^{-1}$ of Cd) by applying non-polluted agriculture soil spiked with a CdCl$_2$·2.5 $H_2O$ solution and an unamended control for comparison. We sprinkled the solution into the soil and left it for a month for equilibration. The garden soil pH was 6.0, with 3.71% organic carbon, 0.25 mg kg$^{-1}$ of total P, 0.12 mg kg$^{-1}$ of total Cd, 1.8 g kg$^{-1}$ of total N and 8.35 cmol kg$^{-1}$ of CEC. The fertilizers in the form of NH$_4$NO$_3$, KH$_2$PO$_4$ and K$_2$SO$_4$ at concentrations of 150 mg kg$^{-1}$ of N, 100 mg kg$^{-1}$ of P$_2$O$_5$ and 150 mg kg$^{-1}$ of K$_2$O, respectively, were applied. Plant seeds were thinned to two parts in

the pot. Plants grew in the greenhouse under natural light and an ambient temperature of 25 ± 3 °C for one month. Each treatment had three replicates.

### 2.3. Determination of Cadmium

Plant tissues were oven-dried at 80 °C for 24 h and then ground with a stainless-steel mill and digested using the $HNO_3$-$HClO_4$ method. The concentration of Cd was measured by an atomic absorption spectrophotometer (Hitachi-Z5000). To evaluate the phytoextraction potential, (I) the bioconcentration factor (BCF), which provides an index of the ability of the plant to accumulate a particular metal with respect to its concentration in the substrate (BCF = Cshoots/Csoil, where C represents the metal concentration), and (II) the translocation factor (TF), which gives the leaf/root Cd concentration [17] and depicts the plant's ability to translocate heavy metals from roots to the harvestable aerial part (TF = [(Cleaf/Croot) × 100]), were used [18].

### 2.4. Statistical Analysis

Due to the tiny amount of biomass in the roots, the root biomass and Cd accumulation in the pot experiment was not determined in this paper. Statistical analyses were performed using the SPSS statistical package. Data were tested for significance level by analysis of variance (ANOVA). The subsequent multiple comparisons among means were examined based on the least significance difference (LSD) test.

## 3. Results

### 3.1. Absorption of Cadmium by B. juncea and B. pekinensis

There was little difference between the shoot's biomass in the six treatments, especially in the treatments of 25 and 50 μM Cd. Chlorosis was accentuated, and necrosis occurred by the day of harvest (data not shown). Cd supply >25 μM reduced the shoot biomass of both plants, although the difference was not significant in *B. juncea* compared to CK (Figure 1). *B. juncea* and *B. pekinensis* showed similar Cd-uptake ability of 149 ug in shoots, except at 10 and 25 μM Cd levels, where *B. juncea* had 37% and 47% higher Cd accumulation than *B. pekinensis*, respectively. Cd uptake in roots followed a similar trend as that in the shoots, where *B. juncea* had higher Cd accumulation than *B. pekinensis* at 10 and 25 μM Cd levels. Both *B. juncea* and *B. pekinensis* showed high Cd-translocation ability in hydroponics. For example, TF was higher than 1 in *B. juncea* in all Cd treatments except for 50 μM Cd (TF = 0.66); meanwhile, TF in *B. pekinensis* ranged from 0.88 to 1.08 (data not shown). In addition, the significant highest Cd content accumulated in shoots occurred in the intermediate Cd concentration levels. For example, *B. juncea* reached the highest Cd content of up to 175 μg Cd in shoots at the 10 μM Cd level, while *B. pekinensis* accumulated 115 μg Cd in shoots at the 5 μM Cd level.

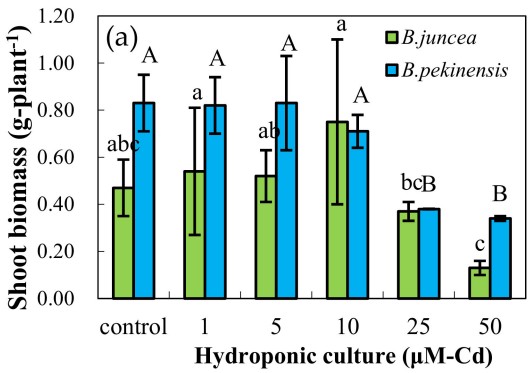
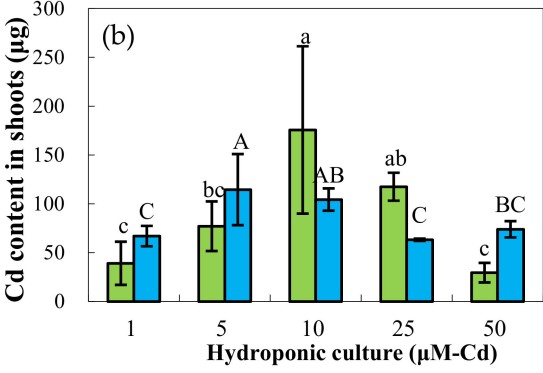

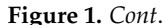

**Figure 1.** *Cont.*

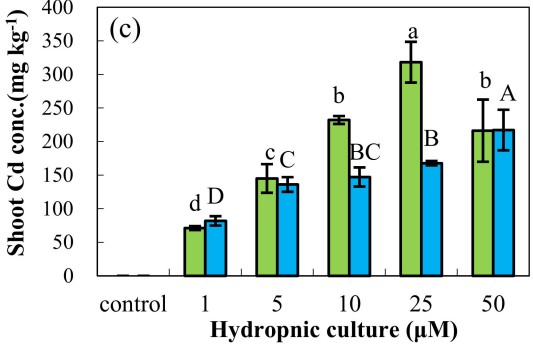

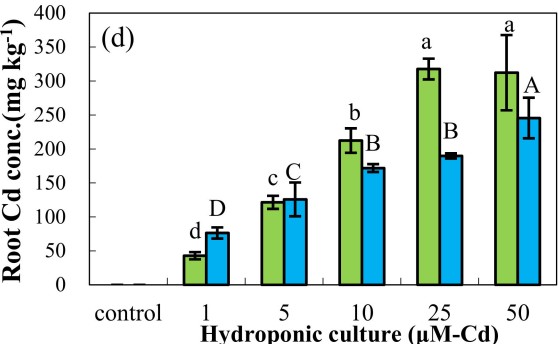

**Figure 1.** Shoot biomass and Cd concentrations in *B. juncea* and *B. pekinensis* grown in hydroponics, including shoot biomass (**a**), the content of Cd in shoots (**b**), the content of Cd in shoot (**c**) and the content of Cd in root (**d**). Different letters in a given column indicate if it is significantly different at $p < 0.05$.

### 3.2. Effects of Cadmium on Plant Growth

The plant shoot's biomass was significantly stimulated by 10 mg kg$^{-1}$ of Cd in soil, which increased by 20% in *B. juncea* and 62% in *B. pekinensis* compared to CK ($p < 0.05$, Figure 2a). However, *B. pekinensis* showed severe phytotoxicity in soil with Cd > 60 mg kg$^{-1}$, and *B. juncea* could not even grow at this concentration. Shoot Cd concentrations of both plants increased when the Cd supply increased significantly, reaching the highest accumulation of 128 mg kg$^{-1}$ in *B. juncea* in 40 mg kg$^{-1}$ of Cd soil and 514 mg kg$^{-1}$ in *B. pekinensis* at 100 mgkg$^{-1}$ of Cd soil ($p < 0.05$, Figure 2b). The content of Cd in the plants was higher than 100 mg kg$^{-1}$ in the treatment of 40 mg kg$^{-1}$.

Under our experimental conditions, the concentration of Cd in the shoots of *B. juncea* ($R^2 = 0.995$) and *B. pekinensis* ($R^2 = 0.970$) were significantly correlated in the soil with <20 mg kg$^{-1}$ of Cd (Figure 3). The bioconcentration factor (BCF) in both plants was much higher than 1 in all Cd treatments, which significantly decreased when the soil Cd concentration increased from 5 to 40 mg kg$^{-1}$. However, the BCF reached 3.20 in *B. juncea* and 4.94 in *B. pekinensis*. The plants even grew in the treatment of 40 mg kg$^{-1}$ of Cd.

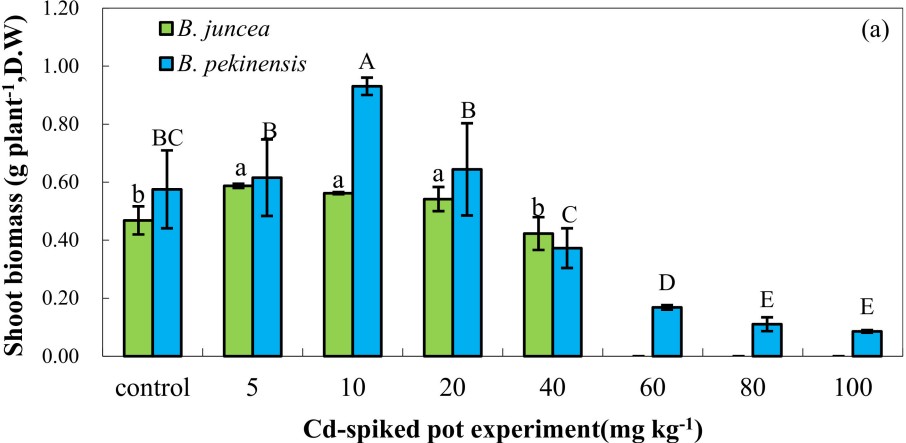

**Figure 2.** *Cont.*

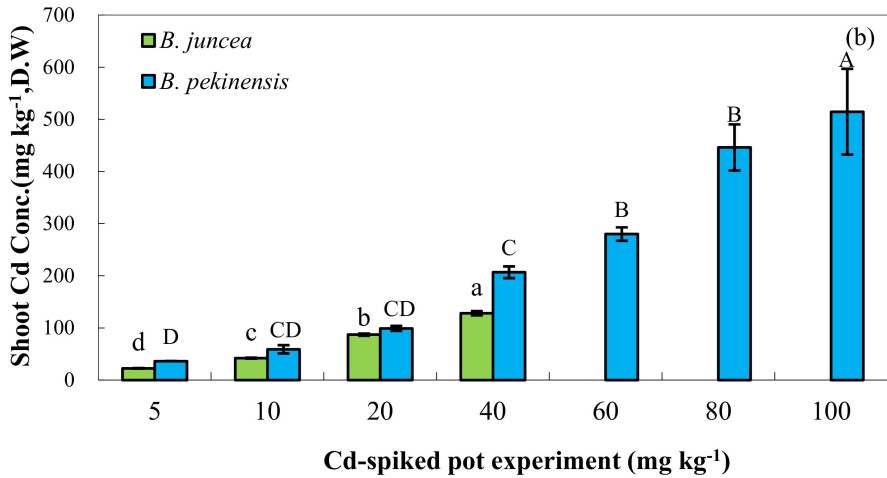

**Figure 2.** Shoot biomass (**a**) and the content of Cd in the shoot (**b**) of *B. juncea* and *B. pekinensis* grown in Cd-spiked soil for 1 month. Different letters in a given column indicate whether it is significantly different at $p < 0.05$.

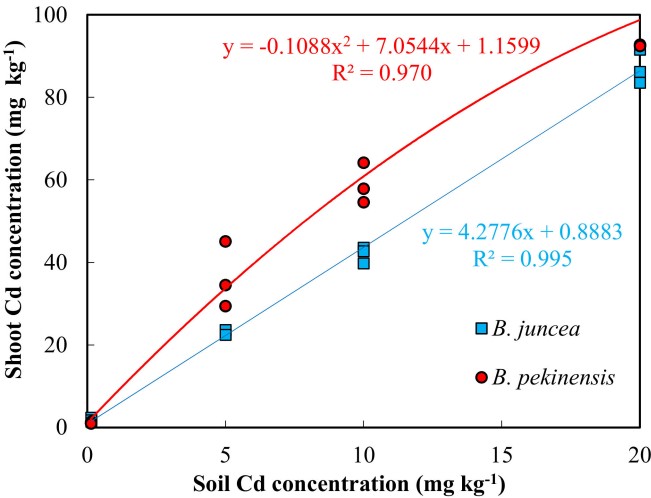

**Figure 3.** Relationship of the content of Cd between the plant shoots and soil.

### 3.3. Evaluation of the Phytoextraction Potential

The hydroponic culture and pot experiment showed that *B. juncea* and *B. pekinensis* both had high Cd accumulation and tolerance at relative low Cd levels. Cd of 5 and 20 mg kg$^{-1}$ were used as target contaminated soil concentrations for *B. juncea* and *B. pekinensis* to still grow without biomass reduction. The possible yield of *B. juncea* is 4–24 t ha$^{-1}$ [19]. It was shown that both plants have a dry biomass production of 4 or 24 t ha$^{-1}$ without fertilization or with optimized agronomic input, respectively. The growth cycle of a plant is about two months, so at least four crops a year can be grown. In the following equation, metal pollution occurred in the soil layers of 20 cm. According to the GB of the Environmental Quality Standard for Soil in China, 0.3 mg kg$^{-1}$ of Cd in soil was used as the target remediation concentration. The phytoremediation potential was evaluated by the following equation, which depended on three variables: plant metal accumulation, soil mass and biomass [20]: % of soil metal removed by one crop = (Plant metal concentration × Biomass)/(Soil meta concentration × Soil mass in the rooting zone) × 100.

For an initial content of 5 mg kg$^{-1}$ of Cd in soil, it would take 13 and 8.5 years to reduce the Cd concentration in the soil to 0.3 mg kg$^{-1}$ for *B. pekinensis* and *B. juncea* with a yield of 24 t ha$^{-1}$, respectively (Figure 4a). In the case of the target 20 mg kg$^{-1}$ of Cd in soil, 15 and 23 years would be required for *B. juncea* and *B. pekinensis* to reduce the Cd

concentration in the soil to 5 mg kg$^{-1}$. However, it would take a much longer time to clean up the soil with a plant biomass of 4 t ha$^{-1}$.

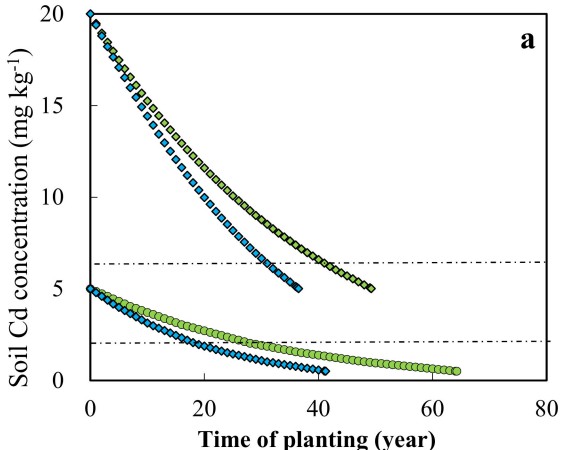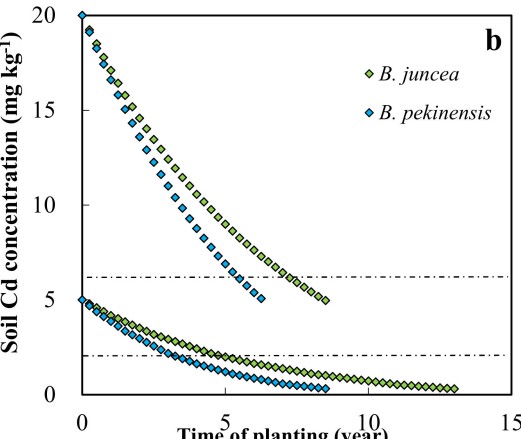

**Figure 4.** Assessment of Cd phytoextraction by *B. juncea* and *B. pekinensis* in Cd-contaminated soil with biomass of 24 (**a**) and 4 t ha$^{-1}$ (**b**). Dashed lines indicate the target for remediation.

## 4. Discussion

### 4.1. Effects of Cadmium on B. juncea and B. pekinensis

*B. juncea* can effectively absorb heavy metals in soil [21]. In the hydroponic culture experiment, both *B. juncea* and *B. pekinensis* showed a high content of Cd accumulation. The content of Cd in the shoots of *B. juncea* and *B. pekinensis* were 232 and 147 mg kg$^{-1}$ at 10 μM of Cd, respectively. These results were similar to Ishikawa et al.'s study [20]. The translocation of Cd to above-ground plant parts has been reported to be mainly determined by liquid flow. [22]. The TF value of both plants was close to 1 with a Cd level <25 μM, which indicated that the internal transport of Cd in *B. juncea* and *B. pekinensis* was unrestricted. However, values obtained from hydroponic culture experiments were different from the field conditions due to the soil's physical and chemical properties [23]. In the case of our pot experiment, the results suggest that *B. pekinensis* and *B. juncea* possess the ability to accumulate and tolerate Cd in soil with Cd levels ranging from 5 to 40 mg kg$^{-1}$ and up to 128 and 207 mg kg$^{-1}$ in the shoots of *B. juncea* and *B. pekinensis* at a 40 mg Cd kg$^{-1}$ level without showing biomass reduction, respectively (Figure 2). In addition, the bioconcentration factor (BCF) was an important factor affecting the efficiency of heavy metal phytoremediation, ranging from 3.2 to 4.6 and 4.94 to 7.27 for *B. juncea* and *B. pekinensis* grown in the treatments of Cd < 40 mg kg$^{-1}$, respectively. The values of BCF obtained in our two plants are much higher than that reported by Ghosh and Singh [24]. In our observation, *B. pekinensis* was more tolerant to Cd than *B. juncea*. For example, *B. juncea* was unable to grow in soil with Cd concentrations >40 mg kg$^{-1}$, though very little soil would exhibit a Cd concentration that high. In addition, the cultivar of *B. pekinensis* (cv. Xiaoza-56) we found can accumulate more Cd in soil than all the cultivars of *B. pekinensis* reported by Liu et al. [25]. It is worth noting that high biomass varieties should not only have the potential to accumulate a large amount of heavy metals but also have the ability to tolerate high concentrations of metals in the soil [26]. Considering the high tolerance to and accumulation of Cd, both *B. juncea* and *B. pekinensis* are good Cd accumulators. However, *B. pekinensis* is a better candidate in phytoremediation of moderate Cd-contaminated soil.

### 4.2. Potential Phytoremediation of Moderate Cd-Contaminated Soil by B. juncea and B. pekinensis

The optimal plant for heavy metal adsorption should not only ensure high tolerance to metal ions but also ensure high yield of the plants [27]. *B. juncea* and *B. pekinensis* are common fast-growing crops in China with high yield production. In this study, these two plants were investigated, and we compared their potential phytoremediation ability for Cd-contaminated soil. There were high Cd-tolerance and high BCF values in both plants,

so we adopted the model from Zhao et al. to evaluate the phytoremediation potential of these two plants [16]. Cd-contaminated soil concentrations of 5 and 20 mg Cd kg$^{-1}$ were selected because they were beneficial to plant growth with a high BCF value. In addition, these two Cd concentrations have been reported as common Cd-contaminated levels in the Dabao mountain area in Guangdong province, China [10]. The model calculation shows that *B. pekinensis* is more effective than *B. juncea* to clean up soil with either an initial concentration of 5 or 20 mg Cd kg$^{-1}$. It may take about five crops to reduce the Cd content in soil from 5 to 3 mg kg$^{-1}$ by *T. caerulescens* with an assumed biomass of 10 t ha$^{-1}$ [16]. In our model calculation, it would require 13 and 8.5 years for *B. juncea* and *B. pekinensis* with an optimized biomass of 24 ha t$^{-1}$ to clean up 5 mg kg$^{-1}$ 261 of Cd in the soil to a concentration of 0.3 mg Cd kg$^{-1}$, respectively. It is clear that using high-biomass plants with a high Cd-tolerance and accumulation is a feasible alternative strategy in soil phytoextraction. Obviously, the production of biomass is an important factor in the efficiency of phytoremediation. For plants with a biomass of 4 t ha$^{-1}$, the time needed to clean up soil from 5 to 0.3 mg kg$^{-1}$ Cd or from 20 to 5 mg kg$^{-1}$ Cd would be 5- or 6-times higher than that of plants with a biomass of 24 t ha$^{-1}$, respectively. It has been reported that improving plant growth with mineral or organic fertilization would not reduce the metal uptake but would significantly increase the removal of metals from available pools [3]. In the future, tillage and weeding can be used to improve the growth environment of crops, and straw mulching can be increased to improve soil fertility. However, we only present the greenhouse experiment and model evaluation here, which cannot refer to the real case in plot phytoextraction. For further field application, appropriate fertilization and agronomic management are needed to optimize the efficiency of phytoextraction in Cd-contaminated soil by *B. juncea* and *B. pekinensis*.

### 5. Conclusions

In our study, the Cd-tolerance and Cd-accumulation of *B. juncea* and *B. pekinensis* (cv. Xiaoza-56) were assessed in hydroponics and pot experiments. It appears that both plants may be suitable for the phytoremediation of moderate Cd-contaminated soil <20 mg kg$^{-1}$ considering the growth cycles, biomass potentials and Cd accumulation. In addition, we found that *B. pekinensis* (cv. Xiaoza-56) is a better candidate in Cd phytoextraction than *B. junceas* since *B. pekinensis* has a stronger adsorption capacity and better tolerance to heavy metals. In the future, field experiments are needed to further develop agronomic practices in order to maximize the Cd removal efficiency in soil.

**Author Contributions:** Z.H. wrote the paper; R.Y. conceived and designed the experiments; R.Y. and X.Z. analyzed the data; R.Q., B.X. and Y.T. performed the experiments and collected the data. All authors have read and agreed to the published version of the manuscript.

**Funding:** This research was funded by the NSFC-Guangdong Joint Foundation of China (No. U0833004), the Guangdong Provincial Natural Science Foundation (No. 06202438) and the Core Project of the National High-tech R&D Program (863 Program, No. 2007AA061001).

**Institutional Review Board Statement:** Not applicable.

**Informed Consent Statement:** Not applicable.

**Data Availability Statement:** The data that support the findings of this study are available upon request from the authors.

**Conflicts of Interest:** The authors declare no conflict of interest.

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
