# Peer review of "Adsorption of Cadmium by Brassica juncea (L.) Czern. and Brassica pekinensis (Lour.) Rupr in Pot Experiment"

_sustainability, doi:10.3390/su14010429_

Round 1
Reviewer 1 Report
Adsorption of Cadmium by Brassica juncea (L.) Czern. and Brassica pekinensis (Lour.) Rupr
General Comments
Overall the present study is useful and will attract a large number of readers/ researchers and provide a detailed description. However, the study required some extensive revision before the final acceptance,
Specific Comments:
Title:
Title is good. But can improve.
Abstract:
Add specification of your work in starting lines
Instead of using the word “we” use this study or some other suitable words.
Revise this sentence
“Which the latter is a better candidate, can be eligible plants”
It’ll be better if you arrange keywords alphabetically.
Introduction:
What is heavy metal? Add some more information about heavy metals effects.
How the heavy metals contamination cause soil and water pollution.
Add some more information about Adsorption process.
Why you choose this metal for Adsorption process.
Revise line 60 and 61.
Why you choose these plants.
Need to add some more data on plants, and their specifications.
Update your data by recently studies and make it better and more efficient.
Material and Methods:
Revise the sentence
“The solution was refreshed twice a week and aerated continually”
Use of suitable words in line 110.
Use of appropriate words in line 124.
“We applied fertilizers”
Revise this line
“Plant seeds were sown 126 and thinned to two plants in individual pot”
Choose of appropriate words
“Plant tissues were oven dried at 80 °C for 24 h, then were ground 131 using a stainless steel mill and digested using the HNO3-HClO4 132 method.”
Revise line 144.
Results and Discussion:
Use of suitable words in line 181.
Need to use appropriate language
“Under our experimental conditions, we observed that”
Revise line 209.
Choose suitable words in 242.
Repeat line 265.
Revise the line, make it correct
“The optimum plant for phytoremediation should not only be 244 able to tolerate and accumulate”
Repeat the line 315 “However, here we only present the”
Conclusion:
The data from your study support this conclusion. Much need to update your study because some too much old studies, you have incorporated replace them with recent and updated knowledge.
Best of Luck
Author Response
Dear Editors and Reviewers: Thank you for your letter and for the reviewers' comments concerning our manuscript entitled “Adsorption of Cadmium by Brassica juncea (L.) Czern. and Brassica pekinensis (Lour.) Rupr in pot experiment” (No. sustainability-1501052). Those comments are all valuable and very helpful for revising and improving our paper, as well as the important guiding significance to our researches. We have studied comments carefully and have made correction which we hope meet with approval. Besides, our current manuscript followed the journal formatting guidelines of sustainability. Revised portion are marked in red throughout the revised manuscript. The main corrections in the paper and the Responses to the reviewer’s comments are as flowing: NOTE: All the Page and Line numbers where revisions were made refer to the Manuscript with marked changes (Manuscript_revised version).docx. The Manuscript_Clean Version was the same version of the Manuscript_revised version with cleaned from all the marks. Responses to the Reviewer 1#’s comments: 1) Overall the present study is useful and will attract a large number of readers/ researchers and provide a detailed description. However, the study required some extensive revision before the final acceptance, Response: Thank you very much for your and comments. Those comments are all valuable and very helpful for revising and improving our paper, as well as the important guiding significance to our researches. We have studied comments carefully and have made correction which we hope meet with approval. Besides, our current manuscript followed the journal formatting guidelines of sustainability. Revised portion are marked in red throughout the revised manuscript. 2) Title is good. But can improve Response: Thank you very much for your comments. We revised it in this part in the latest version of the manuscript. (Page 1, Line1) 3) Instead of using the word “we” use this study or some other suitable words. Response: Thank you very much for your comments. We improved it in the latest version of the manuscript. 4) Revise this sentence“Which the latter is a better candidate, can be eligible plants” Response: Thanks for your comments. We revised it in this part in the latest version of the manuscript. (Page 1, Line 38) 5) It’ll be better if you arrange keywords alphabetically. Response: Thank you very much for your comments. We were rewritten this part in the latest version of the manuscript. (Page 1, Line 41) 6) In introduction,What is heavy metal? Add some more information about heavy metals effects. How the heavy metals contamination cause soil and water pollution. Add some more information about Adsorption process. Why you choose this metal for Adsorption process. Response: Thanks very much for your comments. We revised it and rewritten this part in the latest version of the manuscript. (Page 2, Line 45-49) 7) Revise line 60 and 61. Response: Thanks for your comments. We revised it in this part in the latest version of the manuscript. (Page 2, Line 72-73) 8) Why you choose these plants. Response: Thanks very much for your comments. We added this reason in the latest version of the manuscript. (Page 3, Line 84-86) 9) Need to add some more data on plants, and their specifications. Response: Thanks very much for your comments. We added more data on plants in the latest version of the manuscript. (Page 3, Line 84-86) 10) Update your data by recently studies and make it better and more efficient. Response: Thank you for your comments. We added more data on plants in the latest version of the manuscript. (Page 11, Line 399) 11) Revise the sentence “The solution was refreshed twice a week and aerated continually” Response: Thanks for your comments. We revised it in this part in the latest version of the manuscript. (Page 3, Line 125) 12) Use of suitable words in line 110. Response: Thank you for your comments. We revised it in this part in the latest version of the manuscript. (Page 3, Line 132) 13) Use of appropriate words in line 124. Response: Thank you for your comments. We revised it in the latest version of the manuscript. (Page 4, Line 151) 14) “We applied fertilizers” Revise this line Response: Thanks very much for your comments. We revised it in the latest version of the manuscript. (Page 4, Line 152) 15) “Plant seeds were sown 126 and thinned to two plants in individual pot” Choose of appropriate words Response: Thank you very much for your comments. We revised it in the latest version of the manuscript. (Page 4, Line 155) 16) “Plant tissues were oven dried at 80 °C for 24 h, then were ground 131 using a stainless steel mill and digested using the HNO3-HClO4 132 method.” Revise line 144. Response: Thanks very much for your comments. We revised it in the latest version of the manuscript. (Page 4, Line 160) 17) Use of suitable words in line 181. Response: Thank you very much for your comments. We revised it in the latest version of the manuscript. (Page 5, Line 190) 18) Need to use appropriate language Response: Thank you for your comments. We revised it in the latest version of the manuscript. (Page 5, Line 215) 19) “Under our experimental conditions, we observed that” Revise line 209. 20) Choose suitable words in 242. Response: Thank you for your comments. We revised it in the latest version of the manuscript. (Page 6, Line 229) 21) Repeat line 265. Response: Thanks for your comments. We deleted it in the latest version of the manuscript. (Page 6, Line 245) 22) Revise the line, make it correct “The optimum plant for phytoremediation should not only be 244 able to tolerate and accumulate” Response: Thank you very much for your comments. We revised it in the latest version of the manuscript. (Page 7, Line 257) 23) Repeat the line 315 “However, here we only present the” Response: Thanks for your comments. We deleted it in the latest version of the manuscript. (Page 10, Line 368) 24) The data from your study support this conclusion. Much need to update your study because some too much old studies, you have incorporated replace them with recent and updated knowledge. Response: Thanks for your comments. We added and revised it in the latest version of the manuscript. (Page 10, Line 375-385)

Reviewer 2 Report
Major comments:
- This research presented the use of two Brassica species , which are hyperaccumulator and non-hyperaccumator, for Cd phytoextraction. In addition, this research was carried out with synthetic Cd contaminated soil and water in a hydroponic system. Therefore, the data obtained could be only a draft reference. There were only two experiments of hydroponic and pot experiment with a short time of the plant growth. Fig. 3 was also obtained from the results of Fig.2, and Fig.4 presented the model evaluation, not real data.
- There was no reason behind the experimentation of both hydroponic and pot systems.
- The discussion should be concise and talk about thresholds of Cd that resulted in Cd toxicity and plant physiology, that affected plant growth and the efficiency of Cd phytoextraction.
- Importantly, B. juncea and B. pekinensis are edible plants, therefore, authors should concern and write a remake in the discussion part of this point and also possible phytomanagement.
Minor comment:
- Turnitin found 30% similarity.
- The reference numbers 2, 9, 10 should be rechecked and changed to correct references.
- Writing scientific names must be corrected. Mention in the first time has to be the full name, but later should be only genus and species, and italic.

Author Response
Responses to the Reviewer 2#’s comments:
- This research presented the use of two Brassica species, which are hyperaccumulator and non-hyperaccumator, for Cd phytoextraction. In addition, this research was carried out with synthetic Cd contaminated soil and water in a hydroponic system. Therefore, the data obtained could be only a draft reference. There were only two experiments of hydroponic and pot experiment with a short time of the plant growth. Fig. 3 was also obtained from the results of Fig.2, and Fig.4 presented the model evaluation, not real data.
Response: Thank you very much for your and comments. Those comments are all valuable and very helpful for revising and improving our paper, as well as the important guiding significance to our researches. We have studied comments carefully and have made correction which we hope meet with approval. Besides, our current manuscript followed the journal formatting guidelines of sustainability. Revised portion are marked in red throughout the revised manuscript.
- There was no reason behind the experimentation of both hydroponic and pot systems.
Response: Thanks for your comments. We added the reason and revised it in the latest version of the manuscript. (Page 3, Line 112-113; Page 4, Line 141-142)
- The discussion should be concise and talk about thresholds of Cd that resulted in Cd toxicity and plant physiology, that affected plant growth and the efficiency of Cd phytoextraction.
Response: Thank you very much for your comments. We added and revised it in the latest version of the manuscript.
- Importantly, B. juncea and B. pekinensis are edible plants, therefore, authors should concern and write a remake in the discussion part of this point and also possible phytomanagement.
Response: Thanks very much for your comments. We added the reason and revised it in the latest version of the manuscript.
- Turnit in found 30% similarity.
Response: Thanks very much for your comments. We reduced repetition in the latest version of the manuscript.
- The reference numbers 2, 9, 10 should be rechecked and changed to correct references.
Response: Thank you very much for your comments. We rechecked and changed in the latest version of the manuscript. (Page 11, Line 399)
- Writing scientific names must be corrected. Mention in the first time has to be the full name, but later should be only genus and species, and italic.
Response: Thank you very much for your comments. We added and revised it in the latest version of the manuscript.

Reviewer 3 Report
Manuscript Number: sustainability-1501052-peer-review-v1_comments
Comments
This manuscript has focused on one of the main environmental issues. However, the manuscript needs to address the following minor points before recommending it for publication in the Sustainability.
- Title: Add the word greenhouse or pot experiment to the title of this manuscript
- It is suggested that the term “we” need to be replaced with “this study” in the abstract and whole manuscript
- This study is designed for Cd only, therefore it is suggested to replace the term heavy metal in the last sentence of the abstract with “Cd”
- Line 41-42: These lines need to be supported with the following relevant and recent published literature.
https://www.sciencedirect.com/science/article/pii/S2352186420304703
https://www.sciencedirect.com/science/article/pii/S0026265X19317916
https://www.sciencedirect.com/science/article/pii/S0026265X19305831
- Units need to be uniformly used in whole manuscript as per the journal format.
- Revise heading 2.3 as “determination of cadmium” as this study only focused on the cadmium
- Figure 1: increase the division on y-axis and font size on both axis to improve the visibility of this figure.
- Each term used in the figure need to be explained in the caption
- Figures 2 & 3: increase the division on the y-axis and font size on both axis to improve the visibility of this figure. Each term used in the figure need to be explained in the caption
- Conclusion: this section needs to be strengthened that which of the studied plant species is more accumulator and why?
Author Response
Responses to the Reviewer 3#’s comments:
- This manuscript has focused on one of the main environmental issues. However, the manuscript needs to address the following minor points before recommending it for publication in the Sustainability.
Response: Thank you very much for your and comments. Those comments are all valuable and very helpful for revising and improving our paper, as well as the important guiding significance to our researches. We have studied comments carefully and have made correction which we hope meet with approval. Besides, our current manuscript followed the journal formatting guidelines of sustainability. Revised portion are marked in red throughout the revised manuscript.
- Title: Add the word greenhouse or pot experiment to the title of this manuscript
Response: Thank you very much for your comments. We added and revised the title in the latest version of the manuscript.
- It is suggested that the term “we” need to be replaced with “this study” in the abstract and whole manuscript
Response: Thanks very much for your comments. We revised the title in the latest version of the manuscript.
- This study is designed for Cd only, therefore it is suggested to replace the term heavy metal in the last sentence of the abstract with “Cd”
Response: Thank you for your comments. We revised it in the latest version of the manuscript. (Page 1, Line 38)
- Line 41-42: These lines need to be supported with the following relevant and recent published literature. https://www.sciencedirect.com/science/article/pii/S2352186420304703; https://www.sciencedirect.com/science/article/pii/S0026265X19317916; https://www.sciencedirect.com/science/article/pii/S0026265X19305831
Response: Thank you for your comments. We added the references in the latest version of the manuscript. (Page 11, Line 399)
- Units need to be uniformly used in whole manuscript as per the journal format.
Response: Thank you for your comments. We revised it in the latest version of the manuscript.
- Revise heading 2.3 as “determination of cadmium” as this study only focused on the cadmium
Response: Thank you for your comments. We revised it in the latest version of the manuscript. (Page 4, Line 159)
- Figure 1: increase the division on y-axis and font size on both axis to improve the visibility of this figure.
Response: Thanks for your comments. We revised it in the latest version of the manuscript. (Page 5, Line 206)
- Each term used in the figure need to be explained in the caption
Response: Thanks very much for your comments. We revised it in the latest version of the manuscript.
- Figures 2 & 3: increase the division on the y-axis and font size on both axis to improve the visibility of this figure. Each term used in the figure need to be explained in the caption
Response: Thank you very much for your comments. We added and revised it in the latest version of the manuscript. (Page 6, Line 226-227)
- Conclusion: this section needs to be strengthened that which of the studied plant species is more accumulator and why?
Response: Thank you very much for your comments. We revised the conclusion in the latest version of the manuscript. (Page 6, Line 375-385)
Round 2
Reviewer 1 Report
The authors have improved the article according to the given suggestions. in my opinion, the article may please be accepted for publication.